# Large-scale GWAS of food liking reveals genetic determinants and genetic correlations with distinct neurophysiological traits

Sebastian May-Wilson [1], Nana Matoba [2,3], Kaitlin H. Wade [4,5], Jouke-Jan Hottenga [6], Maria Pina Concas [7], Massimo Mangino [8,9], Eryk J. Grzeszkowiak[1], Cristina Menni [8], Paolo Gasparini[7,10], Nicholas J. Timpson [4,5], Maria G. Veldhuizen[11], Eco de Geus [6,12], James F. Wilson [1,13] & Nicola Pirastu [1,14 ✉]

We present the results of a GWAS of food liking conducted on 161,625 participants from the UK-Biobank. Liking was assessed over 139 specific foods using a 9-point scale. Genetic correlations coupled with structural equation modelling identified a multi-level hierarchical map of food-liking with three main dimensions: "Highly-palatable", "Acquired" and "Low-caloric". The Highly-palatable dimension is genetically uncorrelated from the other two, suggesting that independent processes underlie liking high reward foods. This is confirmed by genetic correlations with MRI brain traits which show with distinct associations. Comparison with the corresponding food consumption traits shows a high genetic correlation, while liking exhibits twice the heritability. GWAS analysis identified 1,401 significant food-liking associations which showed substantial agreement in the direction of effects with 11 independent cohorts. In conclusion, we created a comprehensive map of the genetic determinants and associated neurophysiological factors of food-liking.

[1] Centre for Global Health Research, Usher Institute, University of Edinburgh, Edinburgh, UK. [2] Department of Genetics, University of North Carolina at Chapel Hill, Chapel Hill, NC 27599, USA. [3] UNC Neuroscience Center, University of North Carolina at Chapel Hill, Chapel Hill, NC 27599, USA. [4] Population Health Sciences, Bristol Medical School, University of Bristol, Bristol, UK. [5] Medical Research Council (MRC) Integrative Epidemiology Unit (IEU) at the University of Bristol, Bristol, UK. [6] Dept of Biological Psychology, FGB, Vrije Universiteit Amsterdam, Amsterdam, The Netherlands. [7] Institute for Maternal and Child Health—IRCCS, Burlo Garofolo, Trieste, Italy. [8] Department of Twin Research and Genetic Epidemiology, King's College London, London, UK. [9] NIHR Biomedical Research Centre at Guy's and St Thomas' Foundation Trust, London, UK. [10] Department of Medicine, Surgery and Health Sciences, University of Trieste, Trieste, Italy. [11] Department of Anatomy, Faculty of Medicine, Mersin University, Mersin, Turkey. [12] Amsterdam Public Health research institute, Amsterdam, UMC, The Netherlands. [13] MRC Human Genetics Unit, Institute of Genetics and Cancer, University of Edinburgh, Edinburgh, UK. [14] Human Technopole, Milan, Italy. ✉email: nicola.pirastu@fht.org

Food consumption is one of the most important factors influencing our health and contributes to a large amount of excess mortality in the world[1]. With the near-limitless availability of food in the Western world arising from mass distribution, there has been a shift in factors driving dietary behaviour from merely consuming food that is available to one of choice. For this reason, in parallel to understanding the effect of food consumption on health, there has been an increasing interest in understanding the drivers behind people's food choices. This understanding may then be used to direct consumers toward choices that are more nutritious and thus may reduce the burden of various diseases. Food choice is a complex process which involves many different factors, such as personal preferences, health status, ethical beliefs, and socio-economic context. Rather than measures of preference (or choice), liking of foods reflects the individual hedonic response to foods[2] and is closely related to biology[3–5]. Thus, understanding food-liking may be the first critical step in designing better, more targeted dietary interventions and more acceptable nutritious foods.

Food-liking is a complex trait clearly influenced by genetic inheritance[6], biology, psychology[7], the surrounding environment[8], branding[9], and culture[10]. In particular, twin studies have shown that food preferences are moderately heritable traits, with around 50% of their variance in children being explained by genetic factors plus mostly shared environmental effects[11,12]. In adults, while heritability remains stable, the shared environmental component disappears in favour of the non-shared one (e.g. personal experience)[13–16].

Although several recent GWAS have looked at the genetic variants associated with food consumption[17–20], when it comes to liking, attempts to identify the genetic factors underlying these food-liking traits have focused mostly on candidate gene studies[21] (e.g. genes encoding taste receptors such as TAS2R43 and coffee-liking[22]), with mixed results[23]. More recently, genome-wide approaches have been used to identify several genes related to the liking of different foods in an untargeted manner. For example, genetic variants have now been identified as being associated with the liking of sweet foods[24] or more specific foods[25] such as cilantro/coriander[26]. However, these studies have focused either on specific sensations and tastes or tend to be small in sample size and are underpowered to detect the likely modest effect sizes of common genetic variation on more specific food-liking traits.

Here, we present the results of a genome-wide association study (GWAS) for detailed food- and beverage-liking traits in more than 150,000 participants from the UK Biobank cohort, with replication in up to 26,154 individuals across 11 independent cohorts. Furthermore, we used genetic correlations combined with genomic structural equation modelling to create a multi-level map of the relationships between different food preferences, highlighting three main domains that we define as "Highly palatable", "Low caloric", and "Acquired" foods. We then use a wide range of statistical and bioinformatic analyses to show that the defined dimensions correlate with other objectively measured traits and that the identified loci contain genes which are enriched in relevant tissues and biological functions, indirectly validating the model. Finally, we unravel the pleiotropic effects of many of the identified genetic variants, mapping them to the food-liking traits they influence directly.

## Results

Supplementary Data 1 presents descriptive summary statistics for the food-liking traits.

**Mapping the relationships between food items.** For the first step in our analysis, we aimed to map the relationships between the different food preferences. After running the GWAS on all the questionnaire items, we computed the genetic correlation matrix and compared it with the phenotypic one (Fig. S1). The resemblance between the two correlations was very high (r = 0.91, Supplementary Fig. 1B), but the genetic correlations between the food-liking traits were on average twice as large as the phenotypic correlations, likely due to the high measurement error in the food-liking questionnaire.

Hierarchical factor analysis as described below led to a tree structure model composed of up to 4 levels (Fig. 1A and Supplementary File 1), with three main dimensions of food-liking at the top with the final model composed of 119 questionnaire items out of the initial 144.

The first factor trait included highly energetically rewarding and widely enjoyed foods, such as desserts, meat, and savoury foods, which we named "F-Highly palatable". The second was composed primarily of low-caloric foods, such as vegetables, fruit, and whole grains, which we defined as "F-Low caloric". The third was composed of items for which liking is generally acquired (learned throughout life), such as unsweetened coffee, alcohol, cheese, and strong-tasting vegetables, which we refer to as "F-Acquired". Finally, a fourth minor group was composed of F-sweetened caffeinated drinks.

F-Low-caloric and F-Acquired traits showed a moderately strong genetic correlation ($r_G = 0.59$), while the F-Highly palatable trait was more or less completely independent from either ($r_G$, 0.05 and 0.16, respectively). Finally, the F-Caffeinated Sweet Drinks showed a weak positive correlation with the F-Highly palatable dimension ($r_G = 0.39$) and a weak negative correlation with the F-Acquired and F-Low-caloric groups ($r_G = -0.3$ and $r_G = -0.25$, respectively).

**Genetic correlation with food consumption.** It is not straightforward to validate food-liking as we lack objective measures for it. A common approach is to compare the liking measurements with the corresponding reported consumption. We therefore estimated genetic correlations between food-liking measures and their corresponding reported consumption for all traits where both had been measured. Overall, we detected a very strong genetic correlation between the liking and corresponding consumption traits (Fig. 2, Supplementary Data 5), with all correlation coefficients being >0.7, with the exception of beer ($r_G = 0.4$) and white bread ($r_G = 0.1$). Looking at heritability estimates, the mean SNP heritability for the liking traits (~0.08) was double that for the consumption traits (~0.04), and food-liking always showed higher values. This is with the exception of dried fruit, where there was little evidence of a difference and tea, where heritability was higher for consumption.

**Genetic correlation with other complex traits.** Unfortunately, for a large number of food-liking traits, no comparable food-consumption GWAS exists. This also applies to the higher-order factors of the hierarchical food-liking tree, making direct validation impossible. We can, however, verify if our measurements correlate with other health and socio-economic status traits, as we would expect. For example: we can imagine that liking calorically dense foods will correlate with higher obesity indices and reduced physical activity, while we should see the opposite pattern with low-caloric foods. We therefore estimated genetic correlations between the three higher-order liking factors and the other complex traits included in the ldhub platform.

Genetic correlations with other complex traits (Fig. 3 for selected traits and Supplementary File 2 for full results) showed differences between the three main F-traits. As expected, the F-highly palatable trait showed correlations with higher indices of obesity (higher BMI and body fat percentage), lower socio-

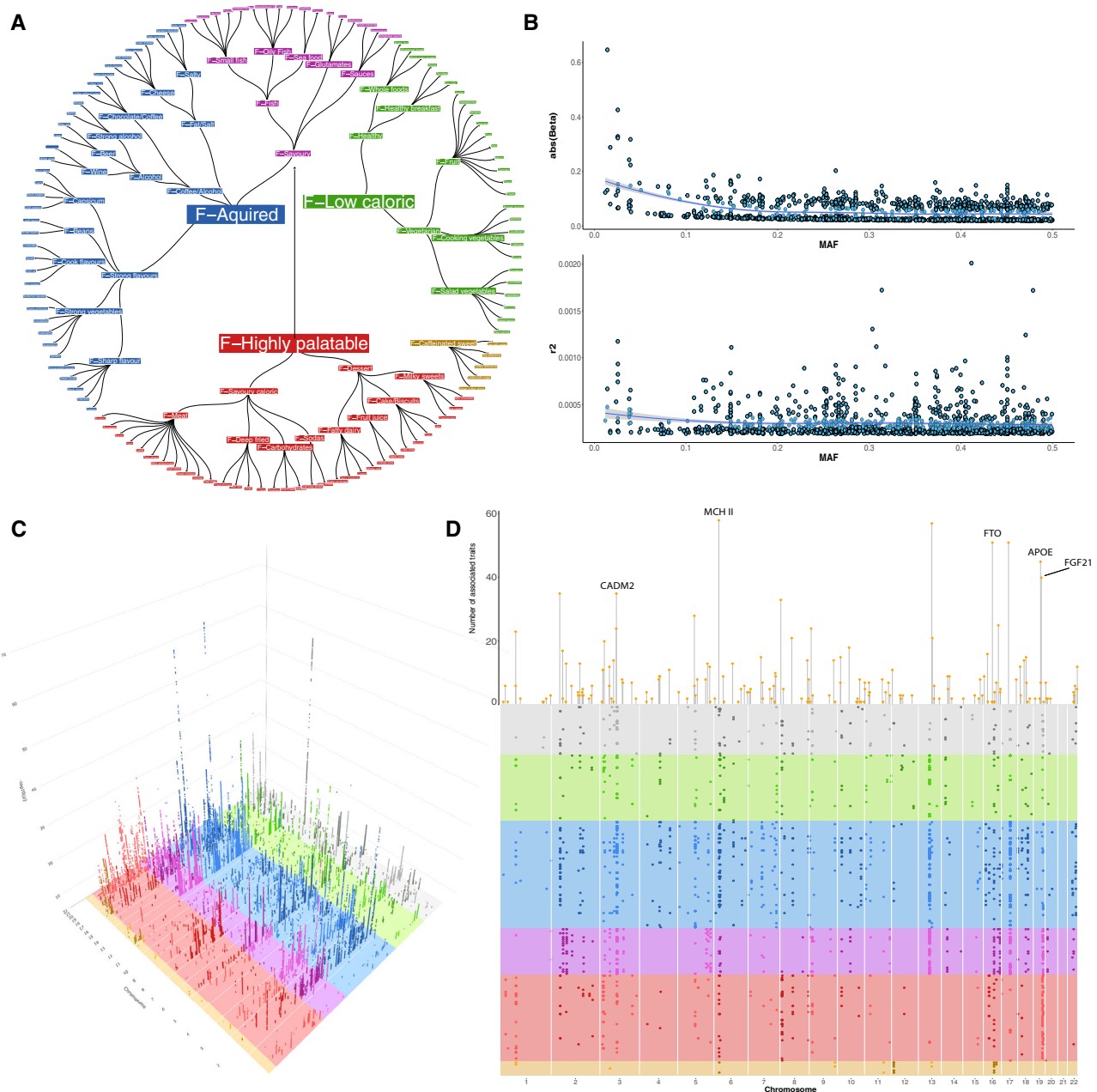

**Fig. 1 Food-liking map and genome-wide association results.** Panel **A** displays the hierarchical model of relationships between liking of different foods. The outermost branches represent the original food-liking traits which were measured with the questionnaire. Colours reflect the membership in one of the four independent dimensions: Red: F-Highly palatable; Blue: F-Acquired; Green: F-Low Caloric; Light brown: F-Caffeinated sweet drinks. F-Savoury foods are coloured purple as they contribute to both F-Highly palatable and F-Acquired Foods. The upper half of panel **B** represents the relationship between the minor allele frequency and effect size. As in most complex traits, there is an inverse relationship between MAF and effect size. The lower panel represents the same SNPs but $r^2$ is reported on the y-axis, showing no relationship between the two measures. The lines represent the trend line with 95% CI. Panel **C** is a 3D Manhattan plot, reporting only SNPs with $p < 5 \times 10^{-8}$. Colours reflect those used in panel **A**. Panel **D** shows a bird's-eye view of the Manhattan plot. Each dot represents the top SNP from each of the sub-loci. The lollipop heights are proportional to the number of traits each locus is associated with.

economic status, and non-sedentary jobs, but lower levels of physical activity. F-Highly palatable was also correlated with higher sodium and creatinine in urine, likely reflective of a diet richer in protein and added salt. The F-Low-caloric trait showed a positive correlation with the use of dietary supplements and higher physical activity, holding non-sedentary jobs, but no relationship with educational attainment was observed. This suggests that people reporting higher liking for the F-Low-caloric trait show a general tendency toward a "healthier" lifestyle

regardless of socio-economic status. This is reflected also by the negative correlation with urinary sodium and creatinine, suggestive of a healthier diet and with a lower body fat percentage. The F-Acquired trait was positively correlated with indices of higher socio-economic status such as years in schooling and a sedentary job, overall healthier blood lipids, lower obesity profile, and higher physical activity, although it also correlated with a higher likelihood of having smoked and higher alcohol consumption.

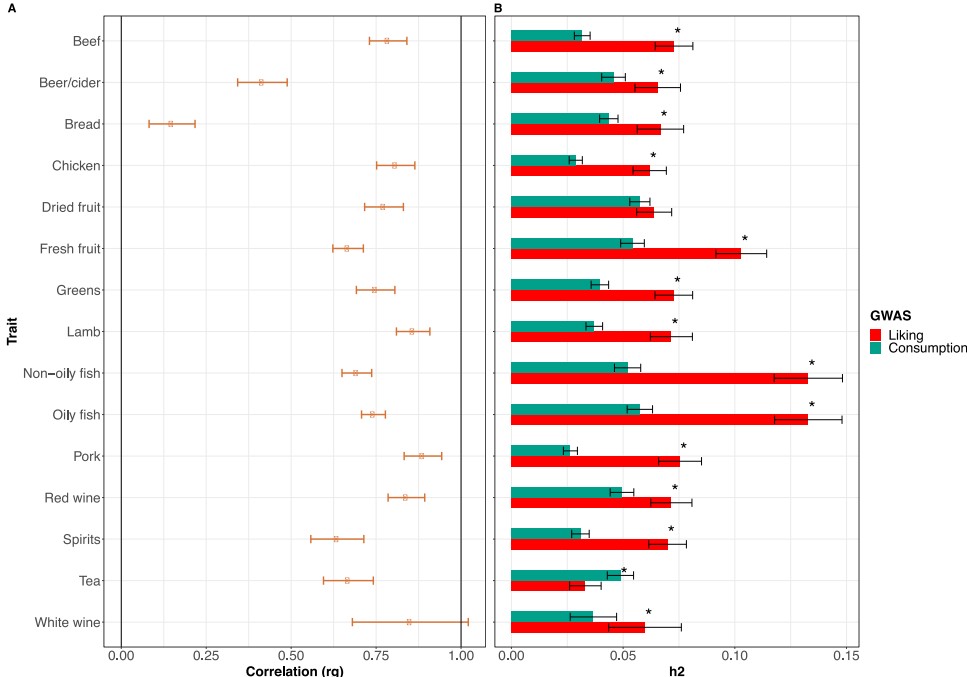

**Fig. 2 Genetic comparison between food-liking and food-consumption traits.** Panel **A** reports the genetic correlations between consumption and liking of the same food for those foods for which both measurements were available, bars represent 95% CI. Panel **B** displays a comparison between SNP heritability of food consumption (green) and liking (red). Bonferroni-corrected significant differences are indicated with a star. Bars represent 95% CI.

**Genetic correlation with brain morphology and connectivity traits.** We further aimed at gathering evidence of the validity of our estimated factors and increasing our understanding of their meaning by looking at genetic correlations with brain imaging traits. Genetic correlations with the brain morphology traits and IC100 rfMRI networks (Fig. 4 and Supplementary Data 13) evidenced clear differences in both types of traits. The morphological associations with the F-Acquired and F-Low-caloric liking dimensions are characterized by negative correlations with cortical thickness in frontal (middle frontal, inferior frontal, and orbital), parietal (intra-parietal and pre-cuneus), and occipital (cuneus, calcarine, and lateral) areas, as well as positive correlations with cortical surface area in the frontal/parietal transition area at the base of the (peri) central sulcus, in the temporal lobe in the fusiform area, and insula. In contrast, the F-Highly palatable liking dimension shows negative correlations with the sub-cortical areas of the basal ganglia in striatal volumes (in the putamen and caudate) and no evident positive correlations.

The connectivity network trait associations are also characterized by the overlap in networks between F-Acquired and F-Low-caloric, which both show (positive and negative) associations with frontal (somato-motor, language), parietal (intra-parietal), temporal (hippocampus, fusiform), and occipital (cuneus) areas. The F-Highly palatable food-liking dimension shows few associations with connectivity networks, and when it does, they are characterized by positive associations with rostral frontal-parietal networks in frontal eye fields and intra-parietal cortex.

Summarizing this, the morphological and network connectivity associations of the food-liking dimensions show parallel effects in the brain, such that both F-Acquired and F-Low-caloric factors show associations with morphology in frontal, parietal, and occipital areas and connectivity in networks involving the same areas, while the F-Highly-palatable dimension shows distinct associations, notably a negative association with the morphology of striatal areas in the basal ganglia.

**GWAS results.** In our GWAS of food-liking, we identified evidence for 1401 genetic associations divided into 173 loci (Fig. 1, Supplementary Data 6). One hundred and forty-three loci out of 173 (~82%), corresponding to 1270 out of 1401 associations, showed correlations with more than one trait, with an average of 8 associated traits per locus. Several loci showed a very high level of pleiotropy. Many of the highly pleiotropic loci (>30 associated traits) map to very well-known genomic regions such as the MHC II locus on chromosome 6 (58 associated traits), the *FTO* locus (51 associated traits) and the *CADM* loci (loci 38 and 39; 59 total associated traits) suggesting that these loci may have a non-specific effect on food-liking.

Replication analysis in up to 26,154 people (median 15,736) from 11 different cohorts was able to replicate 61 (one-tailed $p < 0.05$ and same direction of effect) out of 235 testable associations (26%) (Supplementary Data 9). However, 194 associations corresponding to 82.5% showed consistency in the direction of effect: more than by chance (binomial test $p = 5 \times 10^{-25}$), suggesting that the relatively low number of replicated associations is likely due to a lack of power in the replication cohort.

**Pleiotropy and co-localization.** Given that we had defined loci based only on genomic location we aimed at resolving "true" pleiotropy (i.e. pleiotropy caused by the same genetic variants) using local genetic architectures. For this we performed a co-localization analysis using HyperColoc (Supplementary Data 7, 8). This analysis showed that most traits that were associated with the same locus, also co-localized with the same variants. Within the 143 loci, 138 showed at least one group of traits which co-localized with each other for a total of 203 distinct clusters. 225 of the 1270 associations did not colocalize with any other trait. The only exception was seen at locus 148 which maps to the known chromosome 17 inversion that includes *MAPT* and numerous other genes, where the co-localization process failed.

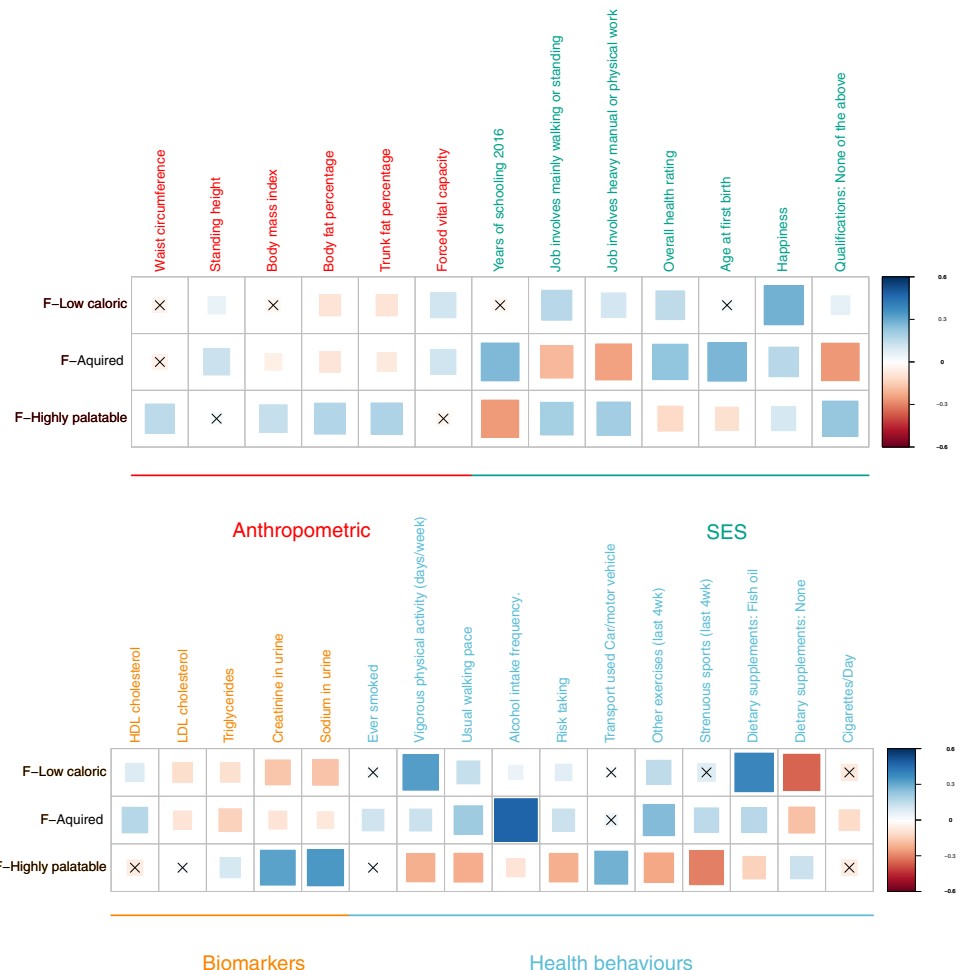

**Fig. 3 Genetic correlation between the three main food-liking factors and other selected complex traits.** X indicates FDR > 0.05. "Qualifications: None of the above" refers to the educational attainment level achieved by the participant and in particular it reflects the lowest qualification possible.

**Distinguishing direct from mediated effects**. As shown by the co-localization analysis, the hierarchical relationships between the food-liking traits give rise to a very high level of pleiotropy. Thus, in order to be able to predict the potential function of the identified genes, it is important to be able to understand at which level of the hierarchical tree of food-liking the variant and locus are primarily associated. If we think of liking fruit, for example, we can imagine that some variants may be associated with all fruits while others may be associated with specific fruits such as apples or oranges. To resolve this issue, we fit the effect of each sentinel SNP onto all nodes of the model at the same time, as outlined in Materials and Methods, and determine if the observed effect was direct or mediated through one of the correlated traits. Of the initial 1261 associations which could be tested within the hierarchical model, only 495 were inferred to be direct effects. As an exemplar case, Fig. 5 shows the effects of this approach on the *ADH1B* locus.

As can be seen, there was strong evidence that the rs1229984 SNP was associated with most alcoholic drinks. However, this SNP had a lesser effect on the stronger alcoholic drinks, suggesting a different weight of alcohol liking, depending on its concentration. After the conditional analysis, only the effect of rs1229984 on F-Alcohol remained unchanged, suggesting that *ADH1B* may exert most of its effect on alcoholic beverages through liking of alcohol in general, although residual effects remain on wine and white wine. Figures for most likely causal SNPs of the 208 association clusters comprising the full model can be found in Supplementary File 3 and Supplementary Data 10.

**Gene prioritization**. Gene prioritization (see Methods for details) allowed us to identify 250 genes as most likely to be causal. Close to half of the associations (43.8%) were intragenic, with roughly 7% of non-synonymous variants and about the same proportion (~6%) of SNPs located either in the 3' or 5' untranslated region. Only ~1% could be explained by synonymous variants.

Rather unsurprisingly, 12 of the prioritized genes encoded either taste (4) or olfactory receptors (8) and highlighted many novel associations. For example, the strongest association we detected was between *OR4K17* and liking of onions (beta = 0.31 on a 9-point scale, $p = 4 \times 10^{-71}$).

Amongst taste receptors, associations were identified only for bitter receptors and all were associated with traits belonging either to the F-Acquired or F-Low-caloric group while none were associated with the F-Highly palatable foods. A similar pattern was observed also for the genes encoding olfactory receptors. Of particular interest are the variants of the *TAS2R38* gene, which were associated with salty foods, alcoholic beverages, horseradish, and grapefruit. This confirmed our previous results which provided evidence for association between this locus and adding salt to food and consuming red wine, but also expanded this finding to other alcoholic beverages[17,27].

Similarly, there were other cases which corroborated and expanded upon previous reports. For example, variants near the *FGF21* gene, which has been previously associated with the consumption of sweet foods[28], were also negatively associated with stronger-tasting foods, especially fish but also eggs, mayonnaise, and fatty foods.

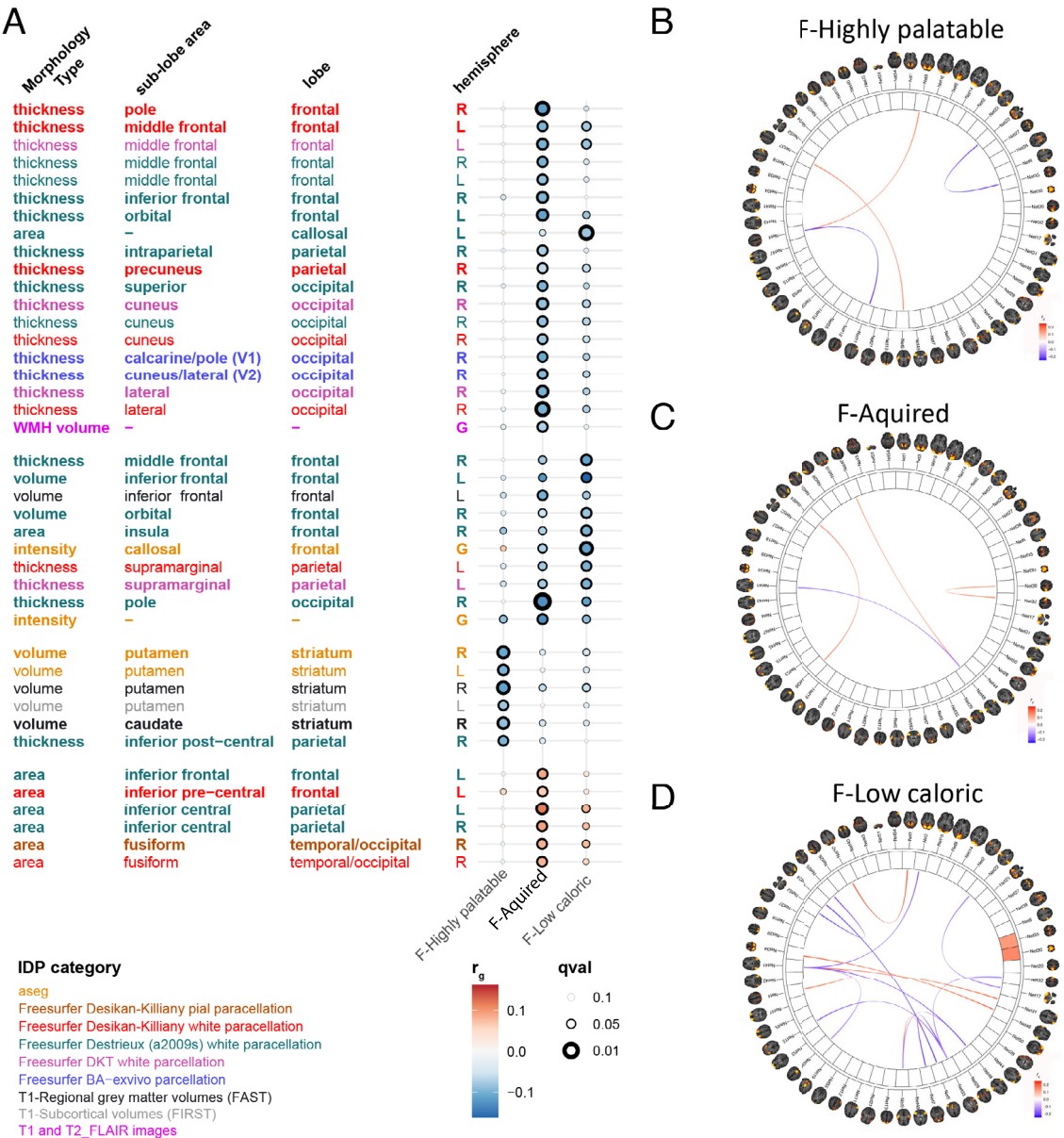

**Fig. 4 Genetic correlations between three main food-liking dimensions and brain MRI traits.** Only traits with q-value < 0.05 have been reported. Panel **A** reports the genetic correlations between the three main liking dimensions and brain MRI morphological traits. Colour reflects the atlas used while size of the dots size is proportional to q-values. Panels **B**–**D** genetic correlations with the ICA100 network traits.

**Tissue and functional enrichment analysis**. Functional enrichment expanding the gene selection to all loci with $p < 5 \times 10^{-8}$ (Supplementary Data 3) resulted in very strong enrichment of cellular components and biological processes related to neurons and specifically to glutamatergic and GABAergic synapses (Fig. 2), both important and well-known modulators of hedonic responses to foods. The tissue enrichment analysis showed evidence for upregulation of one type of tissue only: the brain (Fig. 6; Supplementary Data 11, 12). More specifically, within the brain, primarily sub-cortical regions show upregulation, including the striatal regions of the basal ganglia (putamen, caudate, and nucleus accumbens). Other sub-cortical and paralimbic regions associated with food-reward processing also displayed similar upregulation, such as the hypothalamus, substantia nigra,

amygdala, and hippocampus. Of cortical regions, both the anterior cingulate and frontal cortex show upregulation. These enrichment results converge with the results from the genetic correlation with brain morphology analysis, which showed that the "F-Highly palatable" trait strongly associated with morphology of the striatum in basal ganglia.

## Discussion

In this work, we have examined the genetic bases of food-liking in a comprehensive manner in a Biobank-scale sample. We have shown that it is possible to use genetic correlations to study the relationships between the food traits, highlighting the complexity of these relationships and identifying three main distinct overall

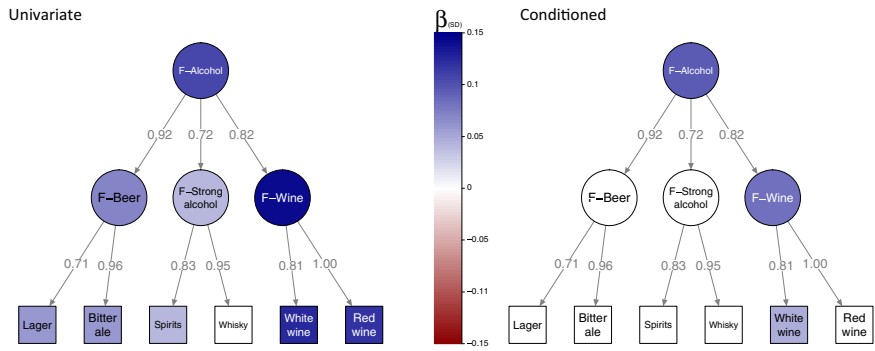

**Fig. 5 Example of univariable vs conditioned analysis of rs1229984 in ADH1B.** The path graph represents the hierarchical model up to the alcohol trait. Numbers over the edges report the standardized loadings. Colour is proportional to effect size. Effect sizes with $p < 1.4 \times 10^{-3}$ (0.05/34 independent traits) have been shrunk to 0.

**Fig. 6 Enrichment analysis of food-liking genes.** Associated biological functions from Gene Ontology (GO) terms and tissue up-regulated genes using the prioritized genes from all loci with $p < 5 \times 10^{-8}$ are shown. The left panels show the summarized significant GO Terms (FDR < 0.05) for cellular component and biological processes, while the right ones report the tissue enrichment using the general tissues (upper panel) and the specific ones (bottom panel).

dimensions while identifying 171 loci involved in 1401 locus-trait associations, most of which have never been described before.

Food-liking has been consistently shown to be a heritable trait in twin studies[11–16], here we have shown that food-liking also has a non-negligible SNP heritability and that it is twice as large as that of food consumption, in line with the idea that food-liking is more influenced by biology than actual behaviour.

The fact that the genetic correlations between liking and food behaviour were relatively high, even when measured ~10 years apart, suggests that the genetic factors underlying these two processes are very similar, while differences likely arise mostly from environmental factors and from the inherent differences between liking and choice. The fact that food-liking is still so strongly correlated to consumption, even if measured later in time, suggests that food-liking is relatively stable through time, at least in adults. Looking at the comparison between genetic and phenotypic correlations amongst the food items, they resemble each other quite closely (r = 0.91), although the genetic correlations are twice as large as the phenotypic correlations. This likely reflects the random measurement error inherent in the use of questionnaires in measuring food-liking and shows that genetic correlations may have advantages in assessing inter-relationships among food-related phenotypes. This strong relationship has been particularly useful in defining our hierarchical model, increasing our ability to identify the underlying dimensions common to multiple foods.

While the current study is not the first to map how liking for different foods is related to each other, this study expands on previous work by having used more than 150 thousand people and covering a wide range of food groups and flavours. In many cases, foods were clustered as expected (e.g. fresh vegetables and fruit) but in other cases have highlighted big differences in foods which are commonly considered as a single group. For example, while the genetic correlation between "cooked vegetables" and "salad vegetables" is very strong (0.79), yet vegetables with stronger tastes such as spinach or asparagus (the "strong vegetables" group) display a much weaker correlation with both cooked and salad vegetables (rG = 0.38 and 0.54, respectively). This is despite the fact that these items would have generally all been categorized together as "vegetables". Our hypothesis-free approach thus captured these previously undescribed differences, which are of great importance in interpreting the results of nutritional studies.

When compared with the results from Vink et al. (2020)[15], our results show a clear resemblance between our first-order traits and those identified through PCA. However, our strategy of using a multi-order hierarchical model allowed the identification of only a few higher-order dimensions, highlighting the minimal correlation between very high reward foods such as sweets, meat, and fried food (the "F-Highly palatable" group) and other lower caloric and stronger taste intensities (F-Low caloric and F-Acquired).

Looking at the genetic correlation with other complex traits, we can see that the F-Highly palatable factor is, as expected, correlated with worse anthropometric and lipid profiles, with signs of a diet rich in protein and salt. The F-Low-caloric and F-Acquired factors show the opposite pattern, both associated with lower indices of obesity and a better blood lipid profile, consistent with a diet lower in salt and protein. When we however look closer, these two factors do show some differences. The F-Acquired factor is associated with higher educational attainment and a sedentary job, likely indices of higher socio-economic status, while for the F-Low caloric we see a different pattern where there is no correlation with educational attainment but a positive one for non-sedentary jobs. Looking at the genetic correlations with the brain MRI morphological traits, while the F-Low-caloric and

F-Acquired foods again show some agreement, the F-Highly palatable foods show none with the other liking dimensions. Strikingly the F-Highly palatable foods correlated only with the morphology of putamen and caudate in the striatum of the basal ganglia. Over-consumption of highly palatable energy-dense foods and adiposity are both associated with downregulation of neural responses in these areas, which is in keeping with previous research[29] When we look at the areas involved with the other two other dimensions, we note they associate with areas involved with sensory responses, identification, and decision making. The specific biological processes that show gene enrichment are mostly related to neural function, especially glutamatergic and GABAergic synapse function. Of all biological tissues tested only the brain shows enrichment and within the brain, specifically subcortical regions including the striatal areas of the putamen and caudate in the basal ganglia stand out. These results converge on neural processes in the brain, and specifically in the basal ganglia, being an important driver of food-liking. Interestingly, several recent studies also highlight the role of the brain in heritable traits that are associated with anthropometric markers for health, like "uncontrolled eating" and physical activity[30,31].

Many studies which have looked at the genetics of food-liking have focused on taste receptors, particularly bitter ones. In this study, we have been able to confirm some of the previous findings such as that of the TAS2R43-46 locus and coffee-liking[22]. For example, we observed a strong association between TAS2R38, responsible for PROP and PTC bitter taste sensations, and both alcoholic beverage and salt liking, confirming our and others' previous results on consumption[17]. We could not, however, replicate the reported associations with any vegetable and, in fact, we found only weak evidence for such an association with broccoli, which was actually in the opposite direction of what would be expected considering previous smaller candidate gene studies. Given that we have looked at a large range of vegetables and the large sample size used, this result questions all previous candidate gene studies that have identified such associations[32]. Similarly, we found little evidence for an association with any of the genes encoding the sweet and umami receptor subunits (TAS1R1-3), again questioning some previous reports in much smaller samples of the association between these genes and sweet-liking[32].

When we look at the genes associated with flavour perception (see Fig S2), namely taste and olfactory receptors, we found that they associate only with the F-Acquired and F-Low-caloric foods and never with the F-Highly palatable foods. It is possible to speculate that this may have an evolutionary meaning, where variants which would lower liking of calorically dense foods such as those in the F-Highly palatable foods would be selected against, while those which increased acceptance of F-Acquired foods which are generally more aversive, would expand one's diet and thus chances of survival. Further, more specific evolutionary genetics studies are needed to test this hypothesis.

Many genes already known to be associated with the consumption of specific foods showed a more complex association pattern, influencing a much broader range of food likings. For example, we have found that the variant rs1229984 within the ADH1B gene was expectedly associated with liking alcoholic beverages, mirroring the results on alcohol consumption. However, when we looked beyond simple genome-wide significance and reduced our p-value threshold, we found that it shows a marginal association with liking sweet foods with a concordant direction of association (Fig. S3). A recent GWAS of sweet-liking[24] conducted in a Japanese cohort where ALDH2, a variant known to be associated with alcohol consumption, is also associated with sweet-liking, but with the opposite effect where the allele associated with higher liking of alcohol is associated with

the lower liking of sweet foods. Both *ADH1B* and *ALDH2* gene products are responsible for metabolizing alcohol in the liver and their association with alcohol consumption is believed to be through the accumulation of acetaldehyde, which gives an unpleasant feeling and thus will reduce alcohol consumption (and like in our case) through conditioned learning. So, although in both populations there is a genetic overlap between alcohol and sweet-liking, this relationship is in opposite directions. These results suggest that the observed association is unlikely to be due to a biological mechanism but further studies involving people who have never consumed alcohol are needed to resolve this issue. Another important example is *FGF21* which has been reported to be associated with the consumption of sugar and protein[19,28]. Previous studies have shown that FGF21 is elevated by low protein and high carbohydrate consumption[33]. Soberg et al. (2017)[34] have previously shown that the rs838133 A allele is associated with lower levels of FGF21 and with higher consumption of sweet foods without an increase in energy intake or obesity. Our results are in line with these studies, with the A allele of rs838133 associated with higher liking of sweet foods; however, when we look at other foods, although liking of some high protein foods (e.g. fish, cheese) is also associated with the A allele, none of the meat traits are associated (Fig. S4). Moreover, we find a much wider range of associated traits, which also include many strong-tasting vegetables and spices, suggesting that the role of FGF21 is indeed to shift liking from sweet to savoury foods, but not necessarily all in the same way.

These examples clearly show how useful our results are in interpreting previous associations, greatly increasing our understanding of the phenomena behind food choices. Our results also highlight the importance of examining food-liking as a whole instead of as sets of distinct sensations, food groups or macronutrients, where the interpretation of the results in one food dimension need to take account of the other factors in order to be properly interpreted. This is particularly important when studying the consequences of food-liking on health status and also when performing Mendelian randomization studies involving food traits.

Another interesting example is the association between a non-synonymous variant in the *GIPR* gene and liking of the foods in the F-Low-caloric group. *GIPR* encodes the receptor of glucose-dependent insulinotropic peptide (GIP), one of two incretins and has been associated with BMI, in particular the A allele is associated with lower BMI[35] and higher liking of low-caloric foods and lower liking of fatty foods such as mayonnaise, cheese and cream (but not fatty meat products such as sausages) (Fig. S5). Amongst many other functions, incretins have been shown to regulate energy metabolism by acting in separate neuronal populations of the central nervous system[36]. GLP-1 and GIP have been shown to regulate food consumption synergistically by acting on the hypothalamic arcuate nucleus increasing neuronal activation and expression of pro-opiomelanocortin[36]. While both hormones are secreted in the presence of sugar, GIP responds also in the presence of free fatty acids[37]. In a recent study, CNS-*Gipr* knockout mice showed lower food intake when exposed to a high-fat diet with smaller meals with consequent lower weight[38]. Our results align very well with this, suggesting that GIPR, similarly to FGF21, is acting through a shift in preferences away from fatty foods and toward lower caloric foods, leading to a lower BMI.

Both these examples point to regulation of food-liking as a possible path through which to regulate food intake quality in order to, for example, help people comply with dietary plans beyond simple regulation of appetite.

The present study has several limitations. The main one is that the study was performed in UK Biobank, which is known to suffer from selection bias. It has been shown before that participants of the UK Biobank are healthier, more educated and older than the general population which may bias the results[39]. This is even more true for people who participate in subsequent questionnaires and thus caution should still be used when interpreting the results, especially regarding genetic correlations[40]. A hint of this potential bias is given by the ranking of the average responses where "unhealthy" more palatable foods rank lower than the "healthy" less palatable ones. Similarly, 'fizzy drinks' and 'tea with sugar' are rated as less liked on average than 'liver.' It is thus possible that some of the loci identified are the result of reverse causality, similarly to what had been observed before for food consumption[17]. Exploring these issues is beyond the scope of this paper and subsequent studies are needed to clarify them.

In conclusion, we have presented an extremely comprehensive GWAS of food-liking in more than 150 thousand individuals. We provided strong evidence that the dimensions of food-liking are not only rooted in culture and familiarity but have an important biological basis, while identifying hundreds of novel associations between genetic variation across the human genome and liking of different foods. This not only greatly increases our knowledge in the field, but opens up numerous paths for further studies aimed at better understanding the processes behind food choice.

## Methods

### Study populations

*UK Biobank.* Analyses were conducted on data collected in the UK Biobank study under project 19655. UK Biobank recruited more than 500,000 people aged 37–73 years from the United Kingdom between 2006 and 2010. The study, participants, and quality control have been described previously[41]. All subjects gave written, informed consent. UK Biobank was approved by the North West Multi-Centre Research Ethics Committee (MREC) and in Scotland, UK Biobank was approved by the Community Health Index Advisory Group (CHIAG). We included only participants who completed the food-liking questionnaire and were of European descent. European descent was defined as being "genomically British" based on genetic principal components or who self-reported as being "Irish", "white" or of "Any other white background". Full details of the genetic information and food-liking phenotypes are presented below.

Genotyping was conducted using the UK Biobank or the UK BiLEVE Axiom Arrays. (Affymetrix, Santa Clara, CA, USA). Further details about imputation, principal components analysis, and QC procedures can be found elsewhere (https://biobank.ctsu.ox.ac.uk/crystal/crystal/docs/genotyping_qc.pdf).

### Food-liking phenotypes.

Food-liking traits were collected through an online questionnaire comprising 152 items, including both foods and drinks plus additional non-food items which captured liking for health-related behaviours such as physical activity. This was administered in 2019 to all UK Biobank participants who had agreed to be recontacted by the study. The questionnaire is an extension of the one previously used in Pallister et al. 2015[13] and Vink et al. 2020[15]. Given that the questionnaire was administered online to participants, pictures were removed, and we used a 9-point Hedonic scale[42], where 1 corresponds to "Extremely dislike" and 9 to "Extremely like". Other options also included "Have never tried it" and "Prefer not to answer". Details of the questionnaire can be found at (https://biobank.ndph.ox.ac.uk/showcase/showcase/docs/foodpref.pdf). Of the 152 items, only the 139 pertaining to food and drink were retained for this specific study, while those which referred to habits such as physical activity or watching TV were not included. Coffee- and tea-liking were each measured twice, with and without sugar. We then defined two additional measures for each. The first was the maximum score given to coffee and tea (coffee max and tea max) to reflect liking for the drink in a preferred way. The other was the difference between the sweetened vs the unsweetened drink to reflect polarization in liking, so higher values meant a higher liking for the sweetened drink while negative numbers reflected a stronger liking for the unsweetened drink. A full list of the food-liking traits used in the study, mean number of participants and standard deviation of responses can be found in Supplementary Data 1.

### Statistical analyses

*GWAS.* Genome-wide association analysis was performed for each of the 144 food-liking traits using the raw reported score. After regressing each food-liking trait with age, sex and the first 10 genetic principal components, array type, and batch, we accounted for genetic relatedness between the participants using GRAMMAR + residuals[43] as estimated in fastGWA[44]. For the analyses, we did not exclude any sample based on their trait value. Finally, GWAS was performed using regscan[45] assuming an additive model on all SNPs with MAF > 0.001. Given the high number

of food-liking traits analyzed and the high correlation between them, to estimate study-wide significance, we first estimated the minimum number of independent components which accounted for at least 95% of the variance overall the traits. This was achieved by estimating the eigen decomposition of the genetic correlation matrix between all the studied food-liking questionnaire items. We estimated that 34 components are sufficient to explain >95% of genetic variance and we thus considered a p-value of $p < 1.47 \times 10^{-9}$ ($5 \times 10^{-8}$ / 34) as the study-wide significance threshold. Given that many loci showed association with multiple traits, we also considered all associations that reached a conventional genome-wide significance threshold ($p < 5 \times 10^{-8}$) if the SNPs were in the same genomic locus as a study-wide significant one.

**Clustering of food-liking items, hierarchical model construction**. To describe the inter-relationships between the food-liking questionnaire items we used hierarchical factor analysis where multiple steps of factor analysis are performed. In our case we first estimated the pairwise genetic correlations between all pairs of the original food-liking items from the questionnaire using the LD-score regression (ldsc) software[46]. As reference we used the SNPs and LD scores referring to HapMap3[47] which are available on the ldsc website. We then performed hierarchical clustering using Ward's D2 method, as implemented in the hclust function of R. We then visually defined the first set of groups that showed a high level of within-group correlation across the individual food-liking items. Next, we estimated the first set of factors, one for each defined group of items. The validity of each of these models was estimated using the GenomicSEM R package[48] and looking at goodness-of-fit metrics, specifically comparative fit index (CFI) > 0.9 and a Standardized Root Mean Square Residual (SRMR) < 0.1. If the model did not have a good fit, we checked whether this could be due to single items and they were removed accordingly. Once the first level of factors was defined, we estimated the effect of each SNP on the factor variable, obtaining for each factor complete GWAS summary statistics. We then estimated genetic correlations between the resulting factors, if any two factors exhibited a genetic correlation larger than 0.9, the items of the two factors were merged together and a new overall factor was estimated. The GWAS on these new factors then became the starting point for building the higher-order factors. This procedure was repeated until we ended up with a hierarchical structure composed of only four highest order factors and up to four levels. To make the results more readable we assigned a label to each of the factors to better interpret what it is capturing (e.g. "Meat" for the factor derived from all the meat-liking traits); however, to keep the difference between observed and derived factor traits, we have prefix the labels with "F-" (e.g. F-Meat).

**Estimation of the effect of each SNP with each factor**. To estimate the effect of each SNP on each of the latent factors we first used GemonicSEM to estimate the loadings of each observed variable onto the latent factor. We then applied the method described in Tsepilov et al. 2020[49]. Briefly, the effect of each SNP on each factor is estimated as the weighted linear combination of the effect of the SNP on each index variable, where the weights are represented by the loadings of each item on the latent factor. This is analogous to using the usergwas function in GenomicSEM, but leads to a large reduction in computing time.

**Comparisons between food-liking and food-consumption traits**. In order to understand how our food-liking measures were related to diet, we performed a genetic correlation analysis between the GWAS of the food frequency and the alcohol consumption questionnaires performed in data, available through the Pan UKBB project website (https://pan.ukbb.broadinstitute.org/). We also compared heritability ($h^2$) estimated using LD-score regression. Heritability comparison and genetic correlations analysis was limited to those traits for which either the exact same item was present in both the food frequency questionnaire and the food-liking questionnaire (e.g. white wine) or items with corresponding and similar items between both questionnaires (e.g. Cheese). These items are listed in Supplementary Data 14.

**Genetic correlations with other complex traits**. Genetic correlations with other complex traits for the three highest order traits were performed using the ldhub web portal (http://ldsc.broadinstitute.org/ldhub/). Given the high number of correlations estimated, we selected a set of 31 traits representative of the socio-economic, anthropometric, blood biochemistry, and health-related behaviour traits, to summarize the results.

**Locus definition and co-localization analysis**. To define the boundaries of each locus, we first selected all SNPs with p-value $< 1 \times 10^{-5}$ and then estimated the distance between each consecutive SNP located on the same chromosome. Two consecutive SNPs were identified as belonging to different loci if they were more than 250 kb apart. This approach allows locus identification based on peak shape rather than a fixed distance from a sentinel SNP. A locus was then considered "significant" if it contained at least one SNP with p-value $< 1.47 \times 10^{-9}$. Loci which showed overlapping boundaries were merged. Finally, to test if the underlying causal SNPs between the merged loci were the same or were just close to each other in the genome, we utilized the HyPrColoc method[50]. Briefly, HyPrColoc tests if a group of traits (e.g. food-liking traits) colocalize and returns the probability of each

SNP in the locus being causal. Moreover, it returns a separate overall regional co-localization probability. We thus divided the positional loci into sub-loci based on the results of this analysis and then used the SNP with the highest probability of being causal for each cluster as the sentinel SNP.

**Meta-analysis and replication**. Replication of the GWAS for the questionnaire items was conducted using up to 26,154 samples coming from 11 different cohorts mostly of European ancestries: ALSPAC, INGI-CARL, INGI-VB, INGI-FVG, CROATIA-Korcula, NTR, Silk Road, the TWINS UK cohort, CROATIA-Vis, and VIKING. Details of each cohort can be found in Supplementary Data 2.

Given that each cohort used related but different questionnaires, the meta-analysis was performed only on the overlapping 54 food-liking traits for which at least 10,000 samples were available.

Given that different cohorts used different liking scales we rescaled the results so that they would reflect a scale going from 0 to 1. Prior meta-analysis QC on the summary stats was performed using EasyQC v 28.3[51].

All traits were meta-analyzed using inverse variance weighting conducted using METAL v 2018-08-28[52].

Given that only a limited number of traits were available for at least ten thousand samples it was possible to attempt replication of only 235 SNP-trait associations out of 1401.

**Gene prioritization**. To define the gene most likely to be responsible for the observed association at each locus, we proceeded with custom prioritization according to the following criteria. We first ran haploR v.4.0.2[53] using $r^2 = 0.8$ as the threshold using the sentinel SNP in each sub-locus. If a SNP was not available within the HaploReg resource, we used the most likely available one. Then, genes were prioritized if the locus met one of the following conditions (in order of importance):

(1) The sentinel SNP is itself or is in strong LD ($r^2 > 0.8$) with a non-synonymous SNP in the gene;
(2) The sentinel SNP is itself or is in strong LD ($r^2 > 0.8$) with a coding SNP in the gene (synonymous or in the untranslated region of the gene);
(3) The top SNP is intronic or is in complete LD with an intronic SNP in the gene;
(4) The top SNP is in strong LD ($r^2 > 0.8$) with an intronic SNP in the gene;
(5) The closest gene.

**Estimating the direct effect of each SNP on specific food-liking and latent factor traits**. One of the aims of this study was to understand which SNPs influence different food-liking traits and if these associations were mediated through some higher-order latent factor or if they were directly influencing the food trait of interest. For example, if we consider alcoholic beverages, we can imagine that some SNPs may influence liking of lower-order food traits such as beer or wine through overall liking of alcohol, or directly influence beer-liking or both. We thus aimed at untangling the direct effect of the SNPs on each food-liking and latent factor trait, from those mediated through other connected traits.

To do this, we used GenomicSEM, which allows fitting the effect of each SNP onto multiple traits at the same time, while considering their relationships. The limitation, however, is that it is not possible to fit the effect of the SNP on all observed variables and the latent variable at the same time, given that the number of observed SNP estimates is less than the parameters we need to estimate.

Therefore, we developed a strategy that enabled us to get all the required estimates. To illustrate this strategy, imagine we have 3 correlated food-liking traits (T1-T3) for which a SNP effect is available and where the common variance can be explained by a latent variable L1 (Fig. 7 Panel A). The first step of our analysis was to estimate the effect of the SNP on the latent variable L1 (Fig. 7 Panel B); to fit the effect of the SNP on all four traits at once to estimate all four parameters, we need to provide at least the same number of observed estimates; however, only three are available. To solve this, we created a new model, where we considered L1 as an observed variable and created a new dummy latent variable (DV) that explained all four traits and that was highly correlated (0.99) with L1. The SNP effect is then fit onto the original three food-liking traits (T1-T3) and the dummy variable such that we could obtain the estimate of the SNP effect on the latent variable and the residuals of the three food-liking traits at the same time.

The described approach is useful to solve simple one-factor models, but it cannot be directly applied to the complex hierarchical model we created, as it would be computationally infeasible. We thus split the hierarchical model of food items into smaller trees, where only one latent variable and its observable food traits were used. In efforts to retain the overall structure, we fixed the loadings of the food-liking traits onto the factor to be the same as those estimated during the construction of the model. Figure 7 panels D-E summarizes this strategy.

For all intermediate order traits, this approach led us to have two different conditional estimates for several factors: one where the latent factor trait was conditioned on the index food traits and another in which it represented the index trait. The estimate which best captured the direct effect was selected by picking the one with the lowest absolute value of Z-score. We hypothesize that if the effect of

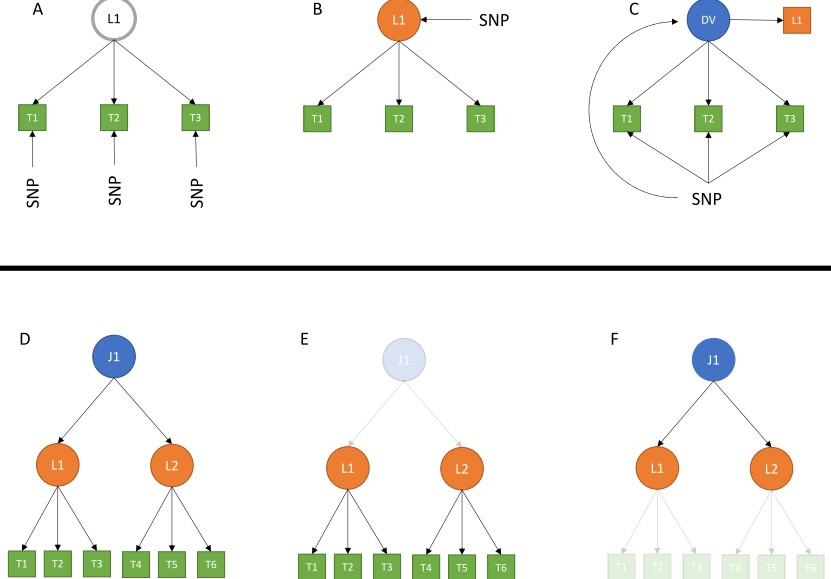

**Fig. 7 Strategy to map loci to specific traits.** Panels **A**–**C** shows how we fit the SNP effect simultaneously on all food-liking traits in the model. We started with the SNP effect on each observed trait participating in the model (**A**). We then used GenomicSEM to estimate the effect of the SNP on the latent variable, L1, based on the observed ones (**B**). We finally used the SNP estimate on L1 as though it were directly observed and created a new dummy latent variable (DV) strongly correlated to L1 (0.99) and fit the SNP effect on LD and all participating food-liking traits at the same time (**C**). Panels **D**–**F** shows the strategy used to fit the multi-order model. The full model (**D**) is split into levels composed of 1 latent variable and its observable.

the SNP is mediated through another trait, conditioning on this trait will lead to a decrease in the effect, and thus the estimate with the smallest effect would correspond to the correct one. Figure 7 panels D, E display a version of this strategy. To test if the conditional SNP estimate was different from the original estimate we used the method from Clogg et al. 1995[54]:

$$ Z = \frac{\beta_1 - \beta_2}{\sqrt{SE_{\beta_1}^2 + SE_{\beta_2}^2}} $$

We considered "direct effect only" SNP/trait effects which showed $p > 0.05$ at this test.

**Functional and tissue enrichment analysis.** For enrichment analysis, we expanded the gene selection to all those genes which were mapped near loci associated with at least one of the food-liking traits at $p < 5 \times 10^{-8}$. Information about the full list of loci can be found in Supplementary Data 3. Tissue enrichment analysis was conducted using FUMA[55] looking at the general and specific GTEx tissues as reference. Gene Ontology term enrichment analysis was conducted using the enrichGO() function from the clusterprofiler R package (3.16.1)[56].

**Correlation with brain MRI traits.** To estimate genetic correlation with brain MRI we first obtained 3,260 GWAS summary statistics on imaging-derived phenotypes (IDP) from multimodal brain imaging (excluded diffusion MRI and ICA25) from Oxford Brain Imaging Genetics Server—BIG40 (https://open.win.ox.ac.uk/ukbiobank/big40/)[57]. These IDPs included morphological traits as well as functional neural response traits. For the morphology measurements cortical thickness, surface area, and volumes were calculated in regional brain areas for various parcellations of the brain (Freesurfer atlases).

Briefly, these areas or brain networks were derived by applying a technique called "group independent component analysis" (ICA) which identifies a prespecified number of networks as independent from each other as possible. For our analysis we used the ICA100 traits which include 55 non-artifact nodes and 1485 edges (between nodes) for a total of 1540 traits. The functional neural response traits included the average neural response over time during a resting-state scan in 55 non-artifact network maps from the ICA100 IDPs (each encompassing multiple regional brain areas), as well as the edges between all 55 ICA maps. The derivation of the ICA100 traits has been described in detail elsewhere[58]. We removed IDPs with low heritability or large uncertainty of heritability estimates ($p < 0.05$), resulting in 2329 IDPs tested for genetic correlations. Genetic correlations were estimated using high-definition likelihood (HDL)[59] to maximize power. Genetic correlations were tested only with the three main dimensions coming from the hierarchical factor analysis. We applied FDR to correct multiple testing on 6987 pairs (the significance threshold was set to $q < 0.05$) (Supplementary Data 4).

**Reporting summary.** Further information on research design is available in the Nature Research Reporting Summary linked to this article.

## Data availability

All GWAS results have been made available through GWAS catalogue accession number GCP000266 Supplementary files 1–3 can be downloaded at: https://osf.io/e43x5/.

## Code availability

Example code can be found at: https://osf.io/e43x5/.

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

## Acknowledgements

We would like to thank all the study participants without whom this study would not have been possible. We would like to thank, Prof Catherine Sudlow for her support to this project, Dr. Jo Holliday for all the work during the final questionnaire definition and roll out. We would further like to thank Dr. Michel Nivard and Dr. Yakov A. Tsepilov for the advice and the great methods they have produced. This research has been conducted using the UK Biobank Resource under Application Number 19655. The Viking Health Study—Shetland (VIKING): DNA extractions and genotyping were performed at the Edinburgh Clinical Research Facility, University of Edinburgh. We would like to acknowledge the invaluable contributions of the research nurses in Shetland, the administrative team in Edinburgh and the people of Shetland. We are extremely grateful to all the families who took part in this study, the midwives for their help in recruiting them, and the whole ALSPAC team, which includes interviewers, computer and laboratory technicians, clerical workers, research scientists, volunteers, managers, receptionists and nurses. Genome-wide association data were generated by Sample Logistics and Genotyping Facilities at Wellcome Sanger Institute and LabCorp (Laboratory Corporation of America) using support from 23andMe. We are very grateful to the municipal administrators of all INGI cohorts, for their collaboration on the project and for logistic support. We would like to thank all participants to this study. J.F.W. acknowledges support from the MRC Human Genetics Unit programme grant, "Quantitative traits in health and disease" (U. MC_UU_00007/10). M.G.V. is supported by the 2232 International Fellowship for Outstanding Researchers Program of TÜBİTAK under award number 118C299. This work was supported by IRCCS Burlo Garofalo of Trieste, funding 5 per mille 2015 senses "Genetics of senses and related diseases" to P.G. Funding was obtained from the Netherlands Organization for Scientific Research (NWO) and The Netherlands Organisation for Health Research and Development (ZonMW) grants 904-61-090, 985-10-002, 904-61-193,480-04-004, 400-05-717, Addiction-31160008, 016-115-035, 400-07-080, Middelgroot-911-09-032, NWO-Groot 480-15-001/674, Center for Medical Systems Biology (CSMB, NWO Genomics), NBIC/BioAssist/RK(2008.024), Biobanking and Biomolecular Resources Research Infrastructure (BBMRI

–NL, 184.021.007 and 184.033.111), X-Omics 184-034-019; Spinozapremie (NWO- 56-464-14192) and KNAW Academy Professor Award (PAH/6635) to D.I.B.; Amsterdam Public Health research institute (former EMGO+); the European Community's Fifth and Seventh Framework Program (FP5- LIFE QUALITY-CT-2002-2006, FP7- HEALTH-F4-2007-2013, grant 01254: GenomEUtwin, grant 01413: ENGAGE); the European Research Council (ERC Starting 284167, ERC Consolidator 771057, ERC Advanced 230374), Rutgers University Cell and DNA Repository (NIMH U24 MH068457-06), the National Institutes of Health (NIH, R01D0042157-01A1, MH081802, DA018673, R01 DK092127-04, Grand Opportunity grants 1RC2 MH089951); the Avera Institute for Human Genetics, Sioux Falls, South Dakota (USA). ALSPAC: The UK Medical Research Council and Wellcome (Grant ref: 102215/2/13/2) and the University of Bristol provide core support for ALS TwinsUK receives funding from the Wellcome Trust (212904/Z/18/Z), Medical Research Council (AIMHY; MR/M016560/1) and European Union (H2020 contract #733100). TwinsUK and M.M. are supported by the National Institute for Health Research (NIHR)-funded BioResource, Clinical Research Facility and Biomedical Research Centre based at Guy's and St Thomas' NHS Foundation Trust in partnership with King's College London. O.M. is supported by Chronic Disease Research Foundation (CDRF). C.M. is funded by the Chronic Disease Research Foundation and by the Medical Research Council (MRC)/British Heart Foundation Ancestry and Biological Informative Markers for Stratification of Hypertension (AIMHY; MR/M016560/1). NJT is a Wellcome Trust Investigator (202802/Z/16/Z), is the PI of the Avon Longitudinal Study of Parents and Children (MRC & WT 217065/Z/19/Z), is supported by the University of Bristol NIHR Biomedical Research Centre (BRC-1215-2001), the MRC Integrative Epidemiology Unit (MC_UU_00011/1) and works within the CRUK Integrative Cancer Epidemiology Programme (C18281/A29019).

## Author contributions

N.P., E.d.G., K.W., N.J.T., J.F.W., and M.G.V. designed the study; C.M., E.d.G., M.M., D.B., J.F.W., K.W., and P.G. provided/collected data; N.P., N.M., S.M.W., M.M., E.J.G., K.W., J.J.H., M.G.V., and M.P.C. analyzed the data; E.D.G., D.B. J.F.W., and P.G. provided funding; N.P., K.W., N.M., M.G.V., J.F.W., N.J.T., and E.D.G. wrote the manuscript. All authors reviewed and provided comments to the text.

## Competing interests

The authors declare no competing interests.
