## [Peer Review File · Nature Communications]

Large-scale genome-wide association study of food liking reveals genetic determinants and genetic correlations with distinct neurophysiological traitsReviewers' Comments:

Reviewer #1:

Remarks to the Author:

Overview. Food liking is a vast and untidy topic, and the goal here is to distill the information without either oversimplifying it or obscuring the main messages. The authors are at least partially successful at meeting these goals. That point made, there are several serious concerns, listed below, about the data and its analysis and interpretation.

To make the most important point first, I have concerns about the validity of the food liking data. In Supplemental Table 1, 'fizzy drinks' and 'tea with sugar' are rated as less liked on average than 'liver.' There is something wrong here that needs to be double-checked!

Speaking of errors, this paper is rife with typos, and there is at least one unformatted reference on line 105. I started to list the typos, but there were so many that by line 137, I had already given up.

One conclusion that is not supported by the data is labeling one of the factors (F-Learning). There is no evidence that the liking for the foods that contribute to this factor is learned; however, these foods are typically not liked by children (e.g., capers). But in my view, it is misleading to call the preferences arising from this developmental shift from more child-like to more adult-like "learning."

There is a sense of mission creep and data sprawl as the authors pull in biological data (gene expression and brain imaging) to help interpret the genetic association results. This effort is not wholly successful. We are left with lots of results that the authors try valiantly to interpret, but the data and results in their entirety do not tell a cohesive story. That point made, the gene ontology classification in Figure 6 was helpful because the tendency is for the genetic associations to occur in genes involved in glutaminergic signaling in the brain is a believable and helpful result. To reduce the sprawl, I would suggest spinning off the brain anatomy data and results into a separate manuscript.

I wonder if it might be easier to understand the data in its entirety if we asked about the nature of the genetic variation (affects one food or many foods) rather than trying to reduce foods into simple factors. I hesitate to suggest additional analysis for a manuscript that is already overfull. Still, I wonder if, rather than reducing the food dimensions, it would be better to start with the observed genotype-phenotype associations and try to conclude those about the underlying biology. For example, based on Table S3, there is only one association with variants in the TAS2R16 gene. In contrast, there are dozens of associations for the CADM2 gene that range from fruits, spices, meats, and red wine. This gene is like a generalist, perhaps contributing to the global experience of food reward, whereas TAS2R16 is a specialist gene, only affecting legumes.

The authors try to understand whether shared food ingredients explain shared genetic effects. For example, they show through conditional analysis that the alcohol in drinks such as beer, wine, and whiskey have a shared genetic association (Figure 5). These analyses were useful, but it would be helpful to draw even broader conclusions: individual foods and drinks that contain the same drug (e.g., alcohol, caffeine) share common genetic determinants.

The most defensible conclusion is that food consumption is less genetically determined than food liking (Figure 3; page 26). The clear message from these data is that many factors wholly unrelated to host biology determine the food eaten on a particular day. People know what they like and reliably report those preferences.

The replication data is a strength of the manuscript.

Minor points.

Figure 2 – panel A – the print is so tiny, and the resolution is so low that I cannot see the labels.

Figure 4 – I did not understand why some boxes are more significant than others; please clarify in the figure caption. Also, I did not understand one of the labels; what is “Qualifications: None of the above”?

The number of variables in the food liking questionnaire, from 152 items down to 139, is not clearly explained. Also, the manipulation of the coffee and tea data seems unnecessarily complicated relative to the knowledge gleaned.

It would be helpful for the authors to make their analysis scripts available with a toy data set so that investigators can better understand the analysis details. This type of sharing is now standard practice, e.g., to put the toy data and analysis scripts on Github.

The resolution for Figure 1S is too low.

Reviewer #2:

Remarks to the Author:

In this paper, Dr. May-Wilson et al generated a comprehensive and data-driven multi-level map of the genetic determinants and associated neurophysiological factors of food liking by conducting a large-scale GWAS for detailed food-and beverage-liking traits in more than 150,000 participants from the UK Biobank, with replication in up to 26,154 participants across 11 independent cohorts.

The authors identified three main food-liking dimensions labeled as: high caloric foods / “Highly palatable”, strong-tasting foods / “Learned” and “Low caloric” foods. By leveraging morphological and functional brain data, they found that these three food-liking dimensions were genetically correlated with non-overlapping, distinct brain areas. The authors also used genomic structural equation modelling to evaluate the direct influence of genetic variants on food-liking traits.

This is a very comprehensive study and the methods are scientifically sound. The paper is clear and well written. My major comments are regarding the review of the literature for previous GWAS for food consumption that do not seem exhaustive, the limited description of the UK Biobank sample used for these analyses, and the absence of limitation section in the discussion.

Below are detailed and additional comments the authors should consider to improve the manuscript.

Major:

- The authors mentioned in the Introduction of the paper that “several recent GWAS have looked at the genetic variants associated with food consumption” and cited three papers, lines 102-103. This does not seem like a comprehensive review of the literature. For instance, a recent large genetic analysis of dietary intake combined data from the CHARGE consortium and the UK Biobank (Merino et al, Nat Hum Behav, 2021). While the reviewer agrees that the outcomes are different, as mentioned in the discussion by the authors “our results also highlight the importance of examining food liking as a whole instead of as sets of distinct sensations/food groups or macronutrients” lines 843-844, the results from this paper also point out to distinct brain regions. It would be interesting to put these results in parallel/compare the findings.

- The description of the UK Biobank sample used in these analyses is pretty succinct/nonexistent. More information is needed for replication purposes and interpretation of the finding. For instance, how European participants were selected? Was it based on self-reported ancestry only or also PCA? Did the authors exclude outliers for each of the outcomes considered? The replication cohort description Table S2 is not practical to read. Maybe the authors could consider moving column I to Supplementary Text.

- There does not seem to be a limitation section in the Discussion and this should be added. The authors could also consider shortening the result summary in the discussion that seem to overlap with

the results section.

- Could the authors clarify if they excluded loci with extremely large effect sizes or extensive long-range LD and traits with low SNP heritability Z scores from the genetic correlation analyses?
- Regarding the replication analysis, the authors mentioned that "194 associations corresponding to 82.5% showed consistency of direction of effect" line 562. However, some P-values in Suppl Table 9 are not significant and thus the effect directions are not really interpretable/reliable. Please rephrase this sentence.
- Could the authors clarify what was their definition for a "at least partially mediated" association? Line 693-694

Minor:

- The methods, results, and discussions section could be shortened by moving some materials in a Supplementary Text.
- P39 lines 702-708: repetitions?
- p5 line 105: "PMID" included instead of reference
- Suggestion to mention the replication in the abstract
- Some Figures are hard to read/interpret, quality should be improved. For instance, Fig 2A & Fig S1 A: blurry, cannot read the food traits
- Fig S1 B: typo x-axis "phenotipic"
- Fig 5: Please consider using lighter blue or white text

We would like to thank the reviewers for the comments which have helped improve our manuscript.

Following point by point responses to their comments.

Reviewer #1 (Remarks to the Author):

Overview. Food liking is a vast and untidy topic, and the goal here is to distill the information without either oversimplifying it or obscuring the main messages. The authors are at least partially successful at meeting these goals. That point made, there are several serious concerns, listed below, about the data and its analysis and interpretation.

To make the most important point first, I have concerns about the validity of the food liking data. In Supplemental Table 1, 'fizzy drinks' and 'tea with sugar' are rated as less liked on average than 'liver.' There is something wrong here that needs to be double-checked!

R. We would like to thank the reviewer for the comment. When we look at these examples, it is good to bear in mind that we are dealing with a population from a specific culture that is between 50-70 years of age and who are on average more educated and healthier than the general population, so it is unsurprising that fizzy drinks and tea with sugar are liked less than liver. We have addressed this source of bias and limitations related to it in the discussion (lines 894-906).

Speaking of errors, this paper is rife with typos, and there is at least one unformatted reference on line 105. I started to list the typos, but there were so many that by line 137, I had already given up.

R. Thanks, we have double checked the language throughout the manuscript and references and made corrections.

One conclusion that is not supported by the data is labeling one of the factors (F-Learning). There is no evidence that the liking for the foods that contribute to this factor is learned; however, these foods are typically not liked by children (e.g., capers). But in my view, it is misleading to call the preferences arising from this developmental shift from more child-like to more adult-like "learning."

R. It is true that very few food likes and dislikes are present at birth and that almost all develop with exposure over time throughout life (Rozin & Vollmecke 1986 (Annu. Rev. Nutr. 1986.6:433-456).. We agree that this group of foods, that we initially labelled "F-Learning" stands out as a group of foods that are generally not liked by children, but mostly by adults. Based on the reviewer's observation, we have changed the label to acquired, consistent with the colloquial description of the liking shifts from child-like to adult-like food, e.g. "It's an acquired taste".

There is a sense of mission creep and data sprawl as the authors pull in biological data (gene expression and brain imaging) to help interpret the genetic association results. This effort is not wholly successful. We are left with lots of results that the authors try valiantly to interpret, but the data and results in their entirety do not tell a cohesive story. That point made, the gene ontology classification in Figure 6 was helpful because the tendency is for the genetic associations to occur in genes involved in glutaminergic signalling in the brain is a believable and helpful result. To reduce the sprawl, I would suggest spinning off the brain anatomy data and results into a separate manuscript.

R. We would like to thank the reviewer for the suggestion; however, we believe that given the novelty of the data and approach the results are necessary to give context to our findings in terms of relationship with biology and other higher-level traits. The measures we derive are the result of statistical elaboration and need to be somehow validated. Given that we don't really

have objective measures of food and drink liking we can see if the result behave at least partly as expected. We have thus shown that:

- 1) they correctly correlate with food consumption measures,
- 2) as expected with behavioural traits genes are enriched in those overexpressed in the brain
- 3) the genes we identify are enriched in pathways related to reward such as the glutaminergic signalling (as noted by the reviewer)
- 4) that highly palatable foods correlated with higher obesity, lower physical activity, and indicators of lower socio-economic status, while the acquired dimension showed an opposite pattern.

These are results which are expected given previous epidemiological work and show that these dimensions behave as we would expect if we were able to measure them directly. This is reinforced by the fact that the two dimensions are independent from each other. The fact that these results align with what is expected also suggest that the rest of the findings are reliable. Moreover, our decision to not spin off the brain anatomy data into a separate manuscript is related to the larger context of our research. Part of the relevance of understanding food-liking drivers lies in finding potential interventions for obesity. Other recent insights have shown that the brain and heritable traits are key in understanding obesity. For example the heritable trait “uncontrolled eating” is associated with obesity and brain systems (Vainik et al. 2019, European Journal of Neuroscience, 50 (3)). Another recent GWAS showed physical activity is heritable, associated with BMI and enrichment in brain tissue and function (Doherty et al. 2018, Nature Communications 9, 5257). This reflects a progressive insight that obesity is driven by heritable traits that are associated with brain function. In agreement, here we show the same convergence of factors related to heritable traits of food liking. To leave out the brain structure and function data would give an incomplete picture of our data-driven discovery of relevant factors. We agree that in the current paper is a lot of data, a range of data analyses and a variety of implications that are challenging to fit into one publication. Therefore, in the revision we have improved cohesion by highlighting the agreement between genetic associations of brain morphology and gene enrichment (Discussion, lines 792-801). In that section we also link our results to other reports that show associations between heritable traits relevant to obesity and the brain.

I wonder if it might be easier to understand the data in its entirety if we asked about the nature of the genetic variation (affects one food or many foods) rather than trying to reduce foods into simple factors. I hesitate to suggest additional analysis for a manuscript that is already overfull. Still, I wonder if, rather than reducing the food dimensions, it would be better to start with the observed genotype-phenotype associations and try to conclude those about the underlying biology. For example, based on Table S3, there is only one association with variants in the TAS2R16 gene. In contrast, there are dozens of associations for the CADM2 gene that range from fruits, spices, meats, and red wine. This gene is like a generalist, perhaps contributing to the global experience of food reward, whereas TAS2R16 is a specialist gene, only affecting legumes.

R. We have reported the data in its entirety. In supplementary file 3, where it is possible to look at which traits are affected by each locus even looking at sub-threshold associations. Describing each of them in detail would be extremely lengthy, although interesting. For example, CADM2 is associated not only to hedonic responses to foods but to most studies' human behaviours including risk taking, number of sexual partners, smoking and being tense. The reader interested in looking at genotype-phenotype associations can thus refer to that section of the paper. The justification for reducing the liking of the 139 foods into factors is that it is unlikely that these liking scores are fully independent, given that sensory profiles and nutrient composition of individual foods overlap with each other. The goal of the paper here was to uncover the factors linking the commonalities driving liking of 139 different foods and to reduce the potentially idiosyncratic ~1400 associations divided in 171 genomic regions with a high level of pleiotropy. Our factor reduction method successfully facilitated this and the observed 3 dimensions associate in a meaningful manner with biology and behavior.

The authors try to understand whether shared food ingredients explain shared genetic effects. For example, they show through conditional analysis that the alcohol in drinks such as beer,

wine, and whiskey have a shared genetic association (Figure 5). These analyses were useful, but it would be helpful to draw even broader conclusions: individual foods and drinks that contain the same drug (e.g., alcohol, caffeine) share common genetic determinants.

R. We would like to thank the reviewer for the comment, and we partly agree with this. In fact although ADH1B association with alcohol beverages is easily explainable through a common ingredient (alcohol), this doesn't apply as a general rule. For example, for the same ADH1B gene the "common ingredient" interpretation does not explain the association with sweet foods or fish. What we wanted to show is that some SNPs influence some foods directly, some through a common higher-level factor shared with multiple foods. This analysis was aimed at distinguishing the two cases. With the ADH1B example we wanted to show that the approach worked in a known case but that each locus shows a different pattern.

The most defensible conclusion is that food consumption is less genetically determined than food liking (Figure 3; page 26). The clear message from these data is that many factors wholly unrelated to host biology determine the food eaten on a particular day. People know what they like and reliably report those preferences.

The replication data is a strength of the manuscript.

Minor points.

Figure 2 – panel A – the print is so tiny, and the resolution is so low that I cannot see the labels.
R. We have added high-resolution figures to the submission.

Figure 4 – I did not understand why some boxes are more significant than others; please clarify in the figure caption. Also, I did not understand one of the labels; what is "Qualifications: None of the above"?

R. We are sorry for the lack of clarity and clarified the label. It refers to "no qualification" in terms of educational attainment and it is thus the lowest grade of education registered.

The number of variables in the food liking questionnaire, from 152 items down to 139, is not clearly explained. Also, the manipulation of the coffee and tea data seems unnecessarily complicated relative to the knowledge gleaned.

R. We reduced the items from 152 to 139 by excluding any items not pertaining to food or drink. We describe this in lines 167-170.
With the additional measures we aimed at capturing the overall coffee liking and if anyone had a particularly sweet tooth in reference to these. Unfortunately we cannot discuss every single finding but still feel that including these measure are helpful results for directing future studies.

It would be helpful for the authors to make their analysis scripts available with a toy data set so that investigators can better understand the analysis details. This type of sharing is now standard practice, e.g., to put the toy data and analysis scripts on Github.

R. We have made the code to produce the overall alcohol factor and the relative conditional analysis available. Overall we have used only established packages so most of the work has been defining the models which are described in the supplementary file 1.

The resolution for Figure 1S is too low.

R. We have provided high resolution figures.

Reviewer #2 (Remarks to the Author):

In this paper, Dr. May-Wilson et al generated a comprehensive and data-driven multi-level map of the genetic determinants and associated neurophysiological factors of food liking by conducting a large-scale GWAS for detailed food-and beverage-liking traits in more than 150,000 participants from the UK Biobank, with replication in up to 26,154 participants across 11 independent cohorts.

The authors identified three main food-liking dimensions labeled as: high caloric foods / “Highly palatable”, strong-tasting foods / “Learned” and “Low caloric” foods. By leveraging morphological and functional brain data, they found that these three food-liking dimensions were genetically correlated with non-overlapping, distinct brain areas. The authors also used genomic structural equation modelling to evaluate the direct influence of genetic variants on food-liking traits.

This is a very comprehensive study and the methods are scientifically sound. The paper is clear and well written. My major comments are regarding the review of the literature for previous GWAS for food consumption that do not seem exhaustive, the limited description of the UK Biobank sample used for these analyses, and the absence of limitation section in the discussion.

Below are detailed and additional comments the authors should consider to improve the manuscript.

Major:

- The authors mentioned in the Introduction of the paper that “several recent GWAS have looked at the genetic variants associated with food consumption” and cited three papers, lines 102-103. This does not seem like a comprehensive review of the literature. For instance, a recent large genetic analysis of dietary intake combined data from the CHARGE consortium and the UK Biobank (Merino et al, Nat Hum Behav, 2021). While the reviewer agrees that the outcomes are different, as mentioned in the discussion by the authors “our results also highlight the importance of examining food liking as a whole instead of as sets of distinct sensations/food groups or macronutrients” lines 843-844, the results from this paper also point out to distinct brain regions. It would be interesting to put these results in parallel/compare the findings.

R. Thanks for the comment. We have added the Merino paper to the introduction although it hadn't come out at the time of submission. We have also added a comment in the discussion comparing the two results. Overall they are similar as we both have found an enrichment in genes expressed in the brain, in our case however we have found a higher number of specific tissues with the two most enriched tissues overlapping the results from the genetic correlations with the morphology.

- The description of the UK Biobank sample used in these analyses is pretty succinct/nonexistent. More information is needed for replication purposes and interpretation of the finding. For instance, how European participants were selected? Was it based on self-reported ancestry only or also PCA? Did the authors exclude outliers for each of the outcomes considered? The replication cohort description Table S2 is not practical to read. Maybe the authors could consider moving column I to Supplementary Text.

R. We have added details about UK biobank and how we defined ancestry to the text. We did not exclude outliers from the traits. There are a few reasons behind this choice. 1) the scale is constrained so this is unlikely to generate issues with the GWAS 2) Despite the fact that this is common practice in GWAS it is not a good one unless it aims at excluding impossible values which are not possible in the case of a constrained Likert scale.

- There does not seem to be a limitation section in the Discussion and this should be added. The authors could also consider shortening the result summary in the discussion that seem to overlap with the results section.

R. We have shortened the summary of the results and added a discussion on the limitations.

- Could the authors clarify if they excluded loci with extremely large effect sizes or extensive long-range LD and traits with low SNP heritability Z scores from the genetic correlation analyses?

R. The munging software removes loci with large effects automatically. We have added this detail to the text. None of the food traits had low heritability while we have specified that brain MRI traits with low heritability or with too large uncertainty were excluded from the analyses.

- Regarding the replication analysis, the authors mentioned that “194 associations corresponding to 82.5% showed consistency of direction of effect” line 562. However, some P-values in Suppl Table 9 are not significant and thus the effect directions are not really interpretable/reliable. Please rephrase this sentence.

R. This is actually quite common in more recent gwas papers. The main problem is that we do not have enough samples in the replication dataset to actually expect to replicate the results, however if the replication betas were distributed by chance we would expect only 50% to be concordant with ours. The fact that this is 82% means that there is a general concordance in the results. Of course this does not mean that all of them are replicated, it is just an overall assessment. We did report how many had also a p-value <0.05 (61).

- Could the authors clarify what was their definition for a “at least partially mediated” association?
Line 693-694

R. We have clarified the text in the manuscript and it now refers only to direct effects.

Minor:

- The methods, results, and discussions section could be shortened by moving some materials in a Supplementary Text.

R. Thanks for the suggestion we will consider it after the paper has reached its final form.

- P39 lines 702-708: repetitions?

- p5 line 105: “PMID” included instead of reference

- Suggestion to mention the replication in the abstract

- Some Figures are hard to read/interpret, quality should be improved. For instance, Fig 2A & Fig S1 A: blurry, cannot read the food traits

- Fig S1 B: typo x-axis “phenotipic”

- Fig 5: Please consider using lighter blue or white text

R. We have corrected each of these minor points.

Reviewers' Comments:

Reviewer #1:

Remarks to the Author:

All comments raised in the previous review have been addressed in this revision

Reviewer #2:

Remarks to the Author:

The authors have addressed the reviews.

Three minor additional comments:

- The authors indicated in their response that "they added a comment in the discussion" to compare their results with Merino et al. As they did not indicate where in the text changes were made and the reference cannot be found in the Discussion section, the reviewer was not able to verify it.
- In Figure 2A, one box does not have any label (red wine?).
- The authors should harmonize spelling of words in the text (for instance: coffeeliking vs. coffee liking)

Here are point by point responses to the reviewer final comments.

Three minor additional comments:

- The authors indicated in their response that “they added a comment in the discussion” to compare their results with Merino et al. As they did not indicate where in the text changes were made and the reference cannot be found in the Discussion section, the reviewer was not able to verify it.

R. We are sorry about this but there was a glitch in the submission system which did not capture the track changes in the pdf.

- In Figure 2A, one box does not have any label (red wine?).

R. We have fixed the figure

- The authors should harmonize spelling of words in the text (for instance: coffeeliking vs. coffee liking)

R. We have checked the manuscript